# Switching to Integrase Inhibitors Unlinked to Weight Increase in Perinatally HIV-Infected Young Adults and Adolescents: A 10-Year Observational Study

**DOI:** 10.3390/microorganisms8060864

**Published:** 2020-06-08

**Authors:** Lucia Taramasso, Antonio Di Biagio, Francesca Bovis, Federica Forlanini, Elena Albani, Rebecka Papaioannu, Vania Giacomet

**Affiliations:** 1Infectious Diseases Unit, Department of Internal Medicine, Fondazione IRCCS Ca’ Granda Ospedale Maggiore Policlinico, University of Milan, 20122 Milan, Italy; lucia.taramasso@hsanmartino.it; 2Infectious Diseases Clinic, Department of Health Sciences (DiSSal), University of Genoa, IRCCS Ospedale Policlinico San Martino, 16132 Genoa, Italy; rebecka.papaioannu@gmail.com; 3Biostatistics Unit, Department of Health Sciences (DiSSal), University of Genoa, 16132 Genoa, Italy; francesca.bovis@gmail.com; 4Pediatric Infectious Disease Unit, ASST Fatebenefratelli-Sacco, University of Milan, 20157 Milan, Italy; federica.forlanini@gmail.com (F.F.); elena.albani1@gmail.com (E.A.); vania.giacomet@unimi.it (V.G.)

**Keywords:** integrase strand transfer inhibitor, weight gain, HIV metabolic complication, perinatal HIV infection

## Abstract

An unexpected increase in weight gain has recently been reported in the course of integrase strand transfer inhibitors (INSTI) treatment. The possibility of this effect in people who are perinatally infected with HIV (PHIV) and thus exposed to lifelong therapy needs to be explored. This is a retrospective multicenter case-control study. Adults with PHIV followed between 2010 and 2019 in two outpatient services in Northern Italy were included if they had at least two weight measures in two successive years of observation. Patients were considered as cases if they were switched to INSTI (INSTI group), or controls if they were never exposed to INSTI (non-INSTI group). The date of the switch in cases was considered to be the baseline (T_0_), while it was randomly selected in controls. Mixed effect models were used to assess the weight changes in INSTI and non-INSTI groups. A total of 66 participants, 50.0% women, 92.4% Caucasian, were included. Median follow-up was 9 years (range 2–10): 4 years (range 1–8) before and 3 (range 1–9) after-T_0_. Mean age at the last study visit was 27.3 (±4.8) years, and mean CD4+ T-cells were 820.8 (±323.6) cells/mm^3^. Forty-five patients were switched to INSTI during the study, while 21 remained in the non-INSTI group. The INSTI group experienced a mean increase (pre-post T_0_) in bodyweight of 0.28 kg/year (95% CI − 0.29; 0.85, *p* = 0.338), while in the non-INSTI group, the mean increase was 0.36 kg/year (95% CI − 0.47; 1.20, *p* = 0.391), without a significant difference between groups (*p* for interaction between time and treatment regimen = 0.868). Among patients on INSTI, the weight gain after T_0_ was higher than pre-T_0_, amounting to +0.28 kg/year (95% CI − 0.29; 0.85), although this difference did not reach significance (*p* = 0.337). PHIV switched to an INSTI-based regimen did not experience an excessive weight gain compared to those who were treated with a non-INSTI based regimen in our cohort.

## 1. Introduction

Several recent studies have highlighted an unexpected adverse effect of new antiretroviral drugs, namely an increase in weight gain among people living with HIV (PLWHIV). Multiple drugs have been blamed to be a possible trigger of this side effect [1,2], and among anchor drugs, integrase strand transfer inhibitors (INSTIs) seem to be those with the highest impact on weight [3,4,5,6]. The weight gain seems higher in naïve, black, female people, with lower baseline weight and a higher pre-treatment HIV-RNA load [1,2,3,4,5]. On the other hand, a weight increase has also been observed after switching strategies in people with virological suppression initiating INSTIs [7,8,9]. The reasons for weight increase as well as the metabolic and cardiovascular impact of the phenomenon remain to be clarified. An excessive weight gain might be especially problematic especially in people infected with HIV perinatally (PHIV), as they are characterized by a pro-inflammatory state [10,11], accelerated ageing [12,13], high prevalence of dyslipidemia [14], and abnormal body fat distribution [15], in a general picture of “metabolic frailty” [16]. In addition, adolescents living with HIV experience multiple forms of stigma, which can further compromise their self-image and weight gain would be even more relevant for them [17].

This population is, by definition, exposed by birth until death to HIV action and, consequently, to lifelong antiretroviral treatment (ART). For these reasons, ART tolerability and cardiovascular safety, as well as any impact on growth, are of particular importance in this setting, not only in the short term after ART initiation, but also after many decades of exposure. However, PHIV are poorly represented in clinical trials and, in many contexts, their ART and switching strategies, are guided by observational studies alone. The possibility of treating them with an INSTI-based regimen is appealing. In fact, it allows overcoming the resistance mutations accumulated in past years of suboptimal ART, to which PHIV have almost invariably been exposed in the early ART era, often limiting possible therapeutic options [18]. Moreover, INSTIs provide the opportunity of reducing the metabolic effects of protease inhibitors (PIs) and older drugs on blood lipids and insulin resistance [16,19,20,21,22] and offer the option of fewer drug and single tablet regimens, an important aspect for improving the adherence in children and adolescents living with HIV [23]. On the other hand, the consequences of possible metabolic effects of these drugs should also be considered, if there are any. In this observational study, we analyse weight changes over 10 years of observation in adults and adolescents, all perinatally infected with HIV, with the aim of evaluating if switching to INSTI has an impact on weight.

## 2. Methods

This is a retrospective, observational, multicentre study, conducted in two referral AIDS centres following vertically HIV-infected children and adolescents in Northern Italy: Policlinico S. Martino Hospital in Genoa and Paediatric Infectious Disease Unit, ASST Fatebenefratelli-Sacco, in Milan. Data for Policlinico S. Martino were retrieved from MedInfo, an online database for anonymous and automatic data collection [24]. Data from the Paediatric Infectious Disease Unit, ASST Fatebenefratelli-Sacco, were retrieved by clinical chart revision. One weight measure per year, or the mean of the weight measures (when more than one weight per year was available), was registered for each year of observation. Data were afterwards merged in a single file for the analysis. All PHIV >18 years old were included in the study if they had at least two weight measures in different years of follow up. Weight measures were retrospectively retrieved for each patient, starting from the first available measure after 18 years of age for males and 16 years for females, to avoid ambiguity related to weight increase as a result of physiological growing [25,26]. We hypothesized that patients switching to or adding an INSTI to ART (INSTI group) would experience greater increases in weight compared to those who remained on non-INSTI ART (non-INSTI group). The criteria for the INSTI group included PHIV (1) who switched from non-INSTI ART, which could have included any nucleoside reverse transcriptase inhibitor (NRTI) backbone along with either adding a non-nucleoside reverse transcriptase inhibitor (NNRTI), protease inhibitor (PI), or entry inhibitor to an INSTI, or adding an INSTI to ongoing ART; (2) at least two weight measures available, one before and one 12 months after the switch and (3) remained on INSTI drugs for at least 12 months following the switch/addition. Patients who performed intra-class switches from one INSTI to another were always considered as part of the INSTI group until the last available follow up or INSTI discontinuation. Criteria for the control (non-INSTI) group included PHIV treated with any drug regimens not including INSTIs and having at least two weight measures. The change of one or more component of the antiretroviral regimen during the follow up did not lead to exclusion from the study in either group. People with missing data or who did not satisfy the above-mentioned criteria were excluded. Pregnant women were excluded from both groups as well.

All data were collected from January 2010 to December 2019. All patients followed in both Policlinico S. Martino Hospital and ASST Fatebenefratelli-Sacco signed an informed consent form in which they agreed to the use of their clinical data in an anonymous form for scientific purposes. The use of the Ligurian HIV Network database for scientific purposes was approved by the Ligurian Ethics Committee (P.R. 2/2013, Date of approval 28 August 2013). The study has been performed in accordance with the ethical standards laid down in the 1964 Declaration of Helsinki and its later amendments, as well as in accordance with Italian national laws.

## 3. Statistical Methods

For clinical and demographic characteristics, descriptive statistics were used, such as mean ± standard deviation (SD), median (range), and frequencies (percentage). Patients’ characteristics in INSTI and non-INSTI groups were compared by chi-square test for categorical variables and Student’s *t*-test for continuous variables. For the INSTI group, the date of the switch was considered as baseline (T_0_). For the non-INSTI group, which never performed the switch to INSTIs, based on the study definition, it was assumed that the weight change was constant over the study period. Therefore, T_0_ was randomly assigned. Furthermore, the two groups have been verified to have a comparable follow-up. Mixed effects models were used to assess weight change per year pre- and post- T_0,_ and to compare the post-T_0_ weight change between the INSTI and non-INSTI group. Models were adjusted for age and weight at baseline and sex. An interaction term between time and treatment regimen was used to assess the effect of the switch to an INSTI regimen on weight change. Additionally, a linear regression model was applied to evaluate the association between birth weight and last available body mass index (BMI) in the study population. SAS 9.3 (Institute Inc., Cary, NC, USA) was used for the computation.

## 4. Results

### 4.1. Weight Gain in the Whole Study Population 

Seventy-eight patients were considered for the present study, of whom twelve were excluded for insufficient weight measures. The remaining 66 patients were included. They had a median follow-up of 9 years (range 2–10 years): 4 years (range 1–8) before-T_0_ and 3 years (range 1–9) after-T_0_. Mean age at the last study visit was 26.7 (SD ± 4.8) years, and the mean CD4+ T-cell count was 820.8 (SD ± 323.6) cells/mm^3^ (Table 1). None of the study participants were in treatment with an INSTI-regimen at the first observation. A mixed effect model adjusted for sex, weight and age at baseline showed a pre-T_0_ weight gain of +0.47 kg/year (95% CI − 0.14; 0.80) and a post-T_0_ weight gain of +0.77 kg/year (95% CI − 0.44; 1.10) (*p* = 0.006 and *p* < 0.0001, respectively), without evidence of a significant change in the rate of weight gain post-T_0_ (0.30 kg/year [95% CI − 0.16; 0.77], *p* = 0.200).

The analysis of weight change between pre- and post-T_0_ by gender showed a slightly increased rate of weight gain post-T_0_ in female patients (+0.46 kg/year (95% CI − 0.16;1.07, *p* = 0.141), while a trend was not seen in male patients (+0.15 kg/year (95% CI − 0.54; 0.84, *p* = 0.660), Table 2. The birth weight did not show any correlation with the last available BMI in the study population (*p* = 0.927).

### 4.2. Comparison of Weight Change in INSTI vs. Non-INSTI-Treated Patients

Among the patients included in the study, 45 switched to an INSTI-based regimen (INSTI group), while 21 were never treated with an ART regimen including INSTI (non-INSTI group). The baseline characteristics of the two groups are shown in Table 1. 

The INSTI group experienced a mean increase (pre-post T_0_) in body weight of 0.28 kg/year (95% CI − 0.29; 0.85, *p* = 0.338), while, in the non-INSTI group, the mean increase was 0.36 kg/year (95% CI − 0.47; 1.20, *p* = 0.391), Table 2. Patients on the INSTI regimen gained slightly less weight compared to the non-INSTI group after T_0_ (−0.09 kg/year), but this difference was not significant (*p* for interaction between time and treatment regimen = 0.868, Figure 1). Six patients had a BMI > 30 and were thus considered obese at the end of the study: three in the INSTI and three in the non-INSTI group.

Analyzing the change in weight by sex, we noticed that male patients on the INSTI regimen seemed to gain slightly less weight compared male patients on a non-INSTI regimen (−0.26 kg/year), while the same trend was not seen in female patients (+0.10 kg/year). Both differences were not significant (*p* for interaction between time and treatment regimen = 0.733 and 0.879, respectively). 

We then adjusted the model for birth weight as well. The model retained 47 patients with available birth weight values (14 never switched, 33 switched to INSTI). Patients who switched to an INSTI regimen seemed to gain more weight compared to those who remained in a non-INSTI regimen (+0.42 kg/year), but this weight increase was not significant (*p* for interaction between time and treatment regimen = 0.522). 

### 4.3. Trend of Weight in INSTI Treated Patients

Finally, we performed a sensitivity analysis only including 45 INSTI-treated patients (Table 3).

These patients seemed to gain slightly more weight after switching to an INSTI regimen compared to the time before the switch (+0.28 kg/year. 95% CI − 0.29; 0.85), but this weight gain was not significant (*p* = 0.337). 

Furthermore, limiting the observation to only at the time following the switch, we did not find a substantial change in weight gain in patients switched for virologic failure (HIV RNA >50 copies/mL the time of INSTI initiation, N = 12) compared to people who switched for other reasons (−0.59 kg/year, [95% CI − 1.77; 0.59], *p* = 0.318).

## 5. Discussion

In this study, we evaluated weight changes over a 10-year period in young adult PHIVs and found that although all patients experienced an increase in weight during the observation, this increase was not linked to INSTI use. A certain rate of weight gain over time, unrelated to ART, could be expected, as even in the general population, young adult men and women have been found to gain 0.55 and 0.52 kg per year [26], respectively, with those who had a rapid and early weight gain during young adulthood being the most likely to be at risk of obesity-related conditions [27]. However, in people living with HIV infection during adulthood, a greater than expected increase of weight was noticed in cases treated with both first line [3,5,28] and switching strategies [7,29] with INSTIs. The reasons for the lack of this same effect among young PHIV might be due to the different characteristics of participants enrolled in this study compared to other contexts. For example, black people were poorly represented, and they seem to be particularly at risk of weight increase in the course of INSTIs [5,28,30]. Moreover, in our study, participants were younger compared to those enrolled in previous studies [7,8,9], and older age is a recognized risk factor for fat and weight increase, at least until elderly age [8,31]. However, studying the weight change at a younger age is still very important, since at as early as 18 years of age, high body mass index can be predictive of increased risk for all-cause mortality and of major ischemic heart disease at old age [32]. Moreover, previous studies highlighted that children infected with HIV have adverse cardiac risk profiles compared with age-paired uninfected controls, presumably as a consequence of both chronic inflammation and exposure to antiretroviral therapy [33,34]. On the other hand, although almost all studies found a weight increase under ART, not all switch studies confirmed the linkage with INSTI use [1,9,35], whose effect in the switch strategies might be less incisive than what was previously reported in naïve patients. Additionally, the weight gain is expected, especially with some INSTIs, such as dolutegravir and bictegravir, and not for all INSTIs [3,5,7,28], with risks differing according to drug combinations [5,36]. However, in our study, due to the small sample size, we did not have the opportunity of sub-group analyses of different INSTIs and different backbones, and this could clearly constitute a limitation. Moreover, we did not find a significant difference in weight increase between the INSTI and non-INSTI groups, but it is possible that the sample size was too small for the data to be significant. In fact, the interaction analysis that we used to investigate the effect modification by the INSTI regimen on the association between time and weight would have required a larger sample size, and the low number of participants in the two groups of the present study may have significantly affected the results. However, the mean increase per year was similar in the two groups of patients, and also to that reported in the general population [27]. Finally, the results that we found in this study in Italy might not be representative of other geographical and cultural contexts where dietary habits as well as attitudes towards physical activity might be very different. Moreover, data on personal choices, health or sociopsychological conditions influencing weight loss or gain were not recorded, limiting the generalizability of these results. With these limitations, our study analyzes an issue that is very important for the future management of PHIV, because many young adolescents and children born with HIV infections now have the possibility to grow old, thanks to the high efficacy of modern ART and drug policies that facilitate the access and supply of drugs, even in limited resource settings [37]. On the contrary, in the past, due to high HIV-related mortality in childhood [38], many PHIVs did not have the chance to reach adulthood and, therefore, there are no historical data on the impact of long-term treatment and on how it is best to treat them over time. This work might have a bearing on potentially greater burden HIV settings, such as Africa, and also in industrialized countries, where mother to child transmission has not been eliminated yet [39].

In conclusion, the weight trajectories that we described in young adult PHIV were similar between INSTI and non-INSTI-treated patients. These results might suggest that INSTI can be used in young adults, who have been perinatally infected with HIV, without expecting excessive weight increase. In the future, it will be increasingly important to join data in multicenter cohort studies in order to confirm these results. 

## Figures and Tables

**Figure 1 microorganisms-08-00864-f001:**
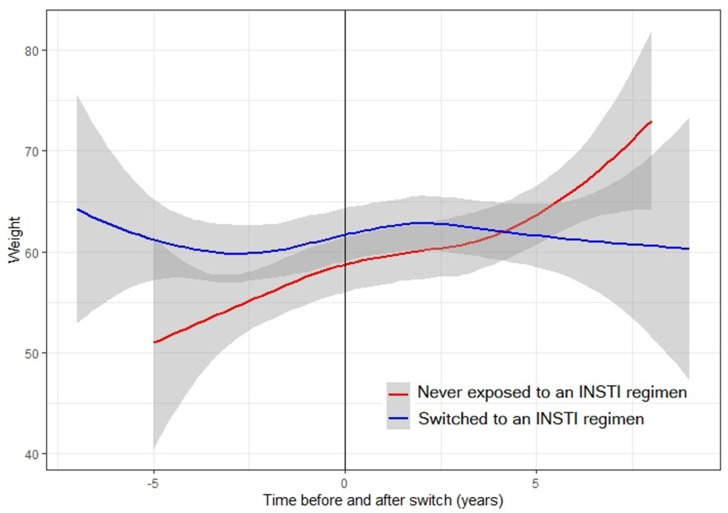
Weight trend during the 10 years follow up in people perinatally infected with HIV, treated with or without integrase strand transfer inhibitors (INSTI)-based antiretroviral regimens.

**Table 1 microorganisms-08-00864-t001:** Baseline demographic and clinical characteristics for the 66 patients analyzed.

Variables	All Patients	INSTI Group	Non-INSTI Group	*p*-Value
**Number of patients (%)**	66 (100)	45 (68.18)	21 (31.82)	
**Mean Age (±SD), years**	26.67 (±4.79)	26.53 (±4.79)	26.95 (±4.90)	0.853
**Male Gender (%)**	33 (50.00)	23 (51.11)	10 (47.62)	0.792
**Mean weight (±SD), kg**	58.65 (12.51)	59.63 (13.66)	56.56 (9.54)	0.711
**Caucasian ethnicity**	61 (92.42)	42 (93.33)	19 (90.48)	0.683
**Mean Born weight (±SD), grams**	2863.40 (±725.64) *N* = 47	2845.15 (±636.62) *N* = 33	2906.43 (±928.83) *N* = 14	0.972
**Mean CD4+T-cell count, cells/mm^3^ (at last clinical evaluation)**	817.68 (±335.21) *N* = 65	770.98 (±348.18) *N* = 44	915.52 (±289.94)	0.059

INSTI: integrase strand transfer inhibitor; SD: standard deviation.

**Table 2 microorganisms-08-00864-t002:** Annual rate of weight change pre and post the time of the switch (T_0_) to integrase strand transfer inhibitor (INSTI). For the non-INSTI group T_0_ was randomly assigned, assuming that the weight change was constant over the study period.

Weight Change in kg/Year	All Patients * (*N* = 66)	Female ** (*N* = 33)	Male ** (*N* = 33)
**Pre-T_0_** **(*p*-value)**	+0.47 (0.14–0.77) (0.006)	+0.52 (0.08–0.95) (0.020)	+0.42 (−0.06–0.91) (0.126)
**Post-T_0_** **(*p*-value)**	+0.77 (0.44–1.10) (<0.0001)	+0.97 (0.54–1.41) (<0.0001)	+0.57 (0.09–1.06) (0.086)
**Pre-post Difference** **(*p*-value)**	+0.30 (−0.16–0.77) (0.200)	+0.46 (−0.16–1.07) (0.141)	+0.15 (−0.54–0.84) (0.660)

* Adjusted for age, sex and weight at baseline; ** adjusted for age and weight at baseline.

**Table 3 microorganisms-08-00864-t003:** Annual rate of weight change pre and post switch to integrase strand transfer inhibitors (INSTI), according to INSTI regimen. For the non-INSTI group the date of the switch (T_0_) was randomly assigned, assuming that the weight change was constant over the study period.

Weight Change in kg/Year	Non-INSTI Group * (*N* = 21)	INSTI Group * (*N* = 45)
**Pre-T_0_** **(*p*-value)**	+0.38 (−0.21–0.97) (0.204)	+0.51 (0.11–0.91) (0.014)
**Post- T_0_** **(*p*-value)**	+0.74 (0.15–1.34) (<.0001)	+0.79 (0.38–1.19) (0.0002)
**Pre-post Difference** **(*p*-value)**	+0.36 (−0.47–1.20) (0.391)	+0.28 (−0.29–0.85) (0.338)

* Adjusted for age, sex and weight at baseline.

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
