# Peer review of "Switching to Integrase Inhibitors Unlinked to Weight Increase in Perinatally HIV-Infected Young Adults and Adolescents: A 10-Year Observational Study"

_microorganisms, 2020, doi:10.3390/microorganisms8060864_

Round 1

Reviewer 1 Report

The authors present an interesting small-sized study regarding INSTI-related weight gain in young adults with HIV infection. In short they found no significant association between switching to INSTI and weight gain. The study results are clear and well-presented. It focuses on a population not often investigated in research studies. I do however have some concerns regarding the presented work. The following issues ought to be addressed before considering the manuscript for publication

Major comments

1. my major concerns is the small number of participants involved in the study. Did the authors investigated whether they had enough statistical power to test their hypotheses (i.e. results of power calculations)?

2. The effect modification by INSTI regimen on the association between time and weight gain has been used throughout the paper to investigate the main outcome. The low number of participants in the two groups may largely affect these results due to "high statistical power" needed in interaction analyses. While I agree that no difference in weight gain is apparent between groups, the authors should comment this limitation.

3. The majority of weight gain after INSTI initiation seems to take place in the first 18 months. Would it be possible to check this assumption in this population?

4. Was the weight at t0 similar in the groups? Would it be possible to control for this confounder in the models?

5. this study focuses on a particular population of PLWH. The presented results may not be applied to other populations (older PLWH), more affected by weight gain. The authors should further address this potential limitation. The last sentence in the conclusion should be reformulated

Minor comments

  1. A table with mixed models results may ease the understanding
  2. stick to either PHIV or PLWHIV
  3. "PHIDs" in the discussion section, typo?

Author Response

The authors present an interesting small-sized study regarding INSTI-related weight gain in young adults with HIV infection. In short they found no significant association between switching to INSTI and weight gain. The study results are clear and well-presented. It focuses on a population not often investigated in research studies. I do however have some concerns regarding the presented work. The following issues ought to be addressed before considering the manuscript for publication

Major comments

  1. my major concerns is the small number of participants involved in the study. Did the authors investigated whether they had enough statistical power to test their hypotheses (i.e. results of power calculations)?

RESPONSE: We thank the reviewer for raising this point and unfortunately, we are aware of the low statistical power due to the small sample size. Children born with HIV infection are a small population in developed countries, where the introduction of ART and the rules of prevention of perinatal HIV transmission have been customary since the late 1990s.  For instance, in the decade 2005-2015 only 79 HIV-1 infected children were born in the whole of Italy (Di Biagio A, Taramasso L, Gustinetti G, et al. Missed opportunities to prevent mother-to-child transmission of HIV in Italy. HIV Med. 2019;20(5):330‐336. doi:10.1111/hiv.12728).Moreover, the peculiarity of this cohort is that it survived, in fact many of children born with HIV infection in the 80s and 90s died due to different reasons, not least the lack of effective ART. Genoa and Milan are two large cities in the north of Italy where the HIV epidemic has hit hardest in the past.

We better highlighted the limitation of the small sample size in the discussion, and we plan to continue to enroll patients over time for confirmatory analyses.

  1. The effect modification by INSTI regimen on the association between time and weight gain has been used throughout the paper to investigate the main outcome. The low number of participants in the two groups may largely affect these results due to "high statistical power" needed in interaction analyses. While I agree that no difference in weight gain is apparent between groups, the authors should comment this limitation.

RESPONSE:  We thank the reviewer for his/her suggestion. We have now addressed this issue in the section disclosing study limitations as follows: “[…] Moreover, we did not find a significant difference in weight increase between INSTI and non-INSTI group, but it is possible that the sample size was too small for the data to be significant. In fact, the interaction analysis that we used to investigate the effect modification by INSTI regimen on the association between time and weight would have required a larger sample size and the low number of participants in the two groups of the present study may have largely affected the results.”

  1. The majority of weight gain after INSTI initiation seems to take place in the first 18 months. Would it be possible to check this assumption in this population?

RESPONSE:  We thank the reviewer for his/her suggestion, and we analyzed the weight gain in the first 12 months after INSTI initiation. The mean weight gain in the patient on INSTI regimen was +0.76 kg compared to the patients on non-INSTI regimen, but the difference was not significant (p=0.653)

  1. Was the weight at t0 similar in the groups? Would it be possible to control for this confounder in the models?

RESPONSE: The mean weight at baseline was 56.56 (±9.54) kg in subjects enrolled in the non-INSTI group and 59.63 (±13.66) kg in the INSTI group (p=0.711). These results are now showed in table 1. As suggested by the reviewer, we adjusted the model for this confounder, and we changed the results accordingly.

  1. this study focuses on a particular population of PLWH. The presented results may not be applied to other populations (older PLWH), more affected by weight gain. The authors should further address this potential limitation. The last sentence in the conclusion should be reformulated

RESPONSE:  We have reformulated the last sentence in accordance with reviewer’s suggestion as follows: “These results might suggest that INSTI can be used in young adults, which have been infected with HIV perinatally, without expecting excessive weight increase.”

Minor comments

  1. A table with mixed models results may ease the understanding

RESPONSE:  According to the reviewer’s suggestion, we added Table 2 and Table 3 to show mixed models results.

Table 2. Annual rate of weight change pre and post the time of the switch (T0) to integrase strand transfer inhibitor (INSTI). For the non-INSTI group T0 was randomly assigned, assuming that the weight change was constant over the study period.

Weight change in kg/year

All patients *

(N=66)

Female**

(N=33)

Male**

(N=33)

Pre-T0

(p-value)

+0.47 (0.14 - 0.77)

(0.006)

+0.52 (0.08-0.95)

(0.020)

+0.42 (-0.06-0.91)

(0.126)

Post-T0

(p-value)

+0.77 (0.44 – 1.10)

(<.0001)

+0.97 (0.54-1.41)

(<.0001)

+0.57 (0.09-1.06)

(0.086)

Pre-post Difference

(p-value)

+0.30 (-0.16 – 0.77)

(0.200)

+0.46 (-0.16 – 1.07)

(0.141)

+0.15 (-0.54 - 0.84)

(0.660)

*Adjusted for age, sex and weight at baseline; **adjusted for age and weight at baseline;

Table 3. Annual rate of weight change pre and post switch to integrase strand transfer inhibitors (INSTI), according to INSTI regimen. For the non-INSTI group the date of the switch (T0) was randomly assigned, assuming that the weight change was constant over the study period.

Weight change in kg/year

Non-INSTI group*

(N=21)

INSTI group*

(N=45)

Pre-T0

(p-value)

+0.38 (-0.21 – 0.97)

(0.204)

+0.51 (0.11-0.91)

(0.014)

Post- T0

(p-value)

+0.74 (0.15 – 1.34)

(<.0001)

+0.79 (0.38-1.19)

(0.0002)

Pre-post Difference

(p-value)

+0.36 (-0.47 – 1.20)

(0.391)

+0.28 (-0.29 – 0.85)

(0.338)

*Adjusted for age, sex and weight at baseline;

  1. stick to either PHIV or PLWHIV

RESPONSE:  In the first version of the paper, we used PHIV to define perinatally infected people, and PLWHIV to indicate, more in general, people living with HIV. We recognize that this generated confusion and used in the new version of the paper only the acronym “PHIV” to define perinatally-infected people.

  1. "PHIDs" in the discussion section, typo?

RESPONSE:  We thank reviewer 1 for giving us the opportunity to edit this typo, that we have now corrected in the revised manuscript.

Reviewer 2 Report

In this work, Dr. Taramasso and colleagues have conducted the analysis on whether Integrase Inhibitors will cause substantial weight gain, as reported by other literatures. Based on their cohort, they observed a moderate weight gain that is not significant (P=0.34). The study is pretty straightforward and is clearly presented. I don’t have any concerns. The only minor question is that why there are only 66 qualified patients from a cohort followed by 10 years?

Author Response

# REVIEWER 2

In this work, Dr. Taramasso and colleagues have conducted the analysis on whether Integrase Inhibitors will cause substantial weight gain, as reported by other literatures. Based on their cohort, they observed a moderate weight gain that is not significant (P=0.34). The study is pretty straightforward and is clearly presented. I don’t have any concerns. The only minor question is that why there are only 66 qualified patients from a cohort followed by 10 years?

RESPONSE:  We thank the reviewer for his/her kind revision and his/her comments. The reason for the small number of subjects enrolled in the present study Is that in Italy mother to child transmission is a quite rare event, thanks to the policy of universal antiretroviral treatment to all people living with HIV, with an enhanced surveillance system (for diagnosis and treatment) during pregnancy. Thus, mother to child transmission is very rare at the time of writing, and many Italian centres follow only sporadic cases of patients perinatally infected with HIV. The two cities where we have conducted the study, Genoa and Milan, are two large centres in the north of Italy where the HIV epidemic has hit hardest in the past. The peculiarity of this cohort is that it survived, in fact many of children born with HIV infection in the 80s and 90s died due to different reasons, not least the lack of effective ART. Then, it is not easy to collect data on larger cohorts of Italian adults infected perinatally (median age 27 years in our study), as the major national and international cohorts of PHIV do not systematically collect data on weight. We better highlighted the limitation of the small sample size in the discussion, and we plan to continue to enroll patients over time for confirmatory analyses.

Reviewer 3 Report

This submitted manuscript investigated the potential effects of integrase strand transfer inhibitors (INSTI) on the weight increase in HIV infected individuals. This is a descriptive human study indicating no association between the usage of INSTI and weight gain. Although this result was interesting, major concerns on this investigation have been raised by the research design.

  • The sample size for each group was pretty much small, so it could be very difficult to reach statistical significance.
  • The weight gain could be attributed by a lot of individual and social factors such as nutrition, education, family income, etc. It would be very difficult to exclude these possibilities.
  • There is no normal control group, i.e., HIV negative group.
  • There is no detailed description on how the weight data was collected. How often? every six months for each patient? How was the health status when weighted?

Author Response

# REVIEWER 3

 This submitted manuscript investigated the potential effects of integrase strand transfer inhibitors (INSTI) on the weight increase in HIV infected individuals. This is a descriptive human study indicating no association between the usage of INSTI and weight gain. Although this result was interesting, major concerns on this investigation have been raised by the research design.

The sample size for each group was pretty much small, so it could be very difficult to reach statistical significance.

RESPONSE: We thank the reviewer for raising this point and unfortunately, we are aware of the low statistical power due to the small sample size. Children born with HIV infection are a small population in developed countries, where the introduction of ART and the rules of prevention of perinatal HIV transmission have been customary since the late 1990s.  For instance, in the decade 2005-2015 only 79 HIV-1 infected children were born in the whole of Italy [Di Biagio A at al., HIV Med. 2019;20(5):330‐336. doi:10.1111/hiv.12728]. Moreover, the peculiarity of this cohort is that it survived, in fact many of children born with HIV infection in the 80s and 90s died due to different reasons, not least the lack of effective ART. Genoa and Milan are two large cities in the north of Italy where the HIV epidemic has hit hardest in the past.

The other peculiarity of this cohort is that, at the time of writing, nobody knows the impact of lifelong effects of INSTI when started in childhood or adolescence, as these drugs did not exist until few years ago and were often not available in resource limited countries, where instead PHIV is frequent. Continue update of the metabolic impact of ART in perinatally infected patients, also after becoming adults, is crucial, but the cohorts of adults with this route of transmission are scarce, exactly because many patients died, discontinued ART or were lost to follow up. Then, we think that also data collected on small cohort deserve attention, also underlying that the power of the study may not be adequate.

We better highlighted the limitation of the small sample size in the discussion, and we plan to continue to enroll patients over time for confirmatory analyses.

The weight gain could be attributed by a lot of individual and social factors such as nutrition, education, family income, etc. It would be very difficult to exclude these possibilities.

RESPONSE:  We thank the reviewer for arising this point, and agree that the lack of these information is an important study limitation. Unfortunately information such as family income or quantity and quality of food intake are difficult to retrieve, and this remains a limitation not only of the present study, but also for most of published study about weight gain in HIV (se for example Venter WDF et al., N Engl J Med. 2019 Aug 29;381(9):803-815. doi: 10.1056/NEJMoa1902824. Sax P et al., Clin Infect Dis. 2019 Oct 14:ciz999. doi: 10.1093/cid/ciz999; Bourgi K et al., J Int AIDS Soc. 2020 Apr;23(4):e25484.  doi: 10.1002/jia2.25484;  Bourgi K et al., Clinical Infectious Diseases. 2020 Mar 17;70(7):1267-1274. doi: 10.1093/cid/ciz407). We discussed this limitation at the end of the manuscript as follows: “Moreover, data on personal choices, health or sociopsychological conditions influencing weight loss or gain were not recorded, limiting the generalizability of these results.”

There is no normal control group, i.e., HIV negative group.

RESPONSE:  We acknowledge that we did not include an HIV-uninfected control group in this work, thus, we still do not know if the weight gain in PHIV is different from the general population. We tried to overcome this limitation by citing literature data in healthy individuals, where weight change seems similar to that found in our study. In fact, reference 26 in the text [Malhotra R, et al. Obesity (Silver Spring). 2013 Sep;21(9):1923–34] cites a work in which the annual weight gain averaged 0.53 kg across a sample of 10,038 young adult men and women (non-HIV infected), similar to what found in our study (pre-T0 increase of 0.47 kg/years and post-T0 increase of 0.77 kg/years). However, the research question of our study was focused on the impact on a particular class of antiretrovirals (i.e. INSTIs) that has been blamed to cause weight gain compared to others. For this reason, we think that choosing INSTI-treated PHIV as cases and non-INSTI treated PHIV as controls may also be acceptable.

There is no detailed description on how the weight data was collected. How often? every six months for each patient? How was the health status when weighted?

RESPONSE:  Weight is usually collected in course of each clinical visit, that is scheduled every 6 months for patients with stable condition, or oftener in case of virological, immunological or comorbidity issues. As the study was retrospective, we did not choose a standard schedule for weight collection, but considered eligible for the study all patients who had at least one weight registered for each year of observation. People with less than 1 measure /year were excluded, as well as pregnant female patients. This information has been detailed in methods: “One weight measure per year, or the mean of the weight measures (when more than one weight per year was available), was registered for each year of observation.”  Overall, twelve patients were excluded for insufficient weight measures (this has been declared at the beginning of the Results section).

As regarding the health status of the enrolled subjects, we agree with the reviewer that it could influence the study results. In this work we only considered the HIV RNA load at the time of the switch and the baseline weight as proxy of the health status of patients, but a multitude of other factors influencing the global health of each participants have not been considered. We have now added this issue among the study limitations as follows: “Moreover, data on personal choices, health or sociopsychological conditions influencing weight loss or gain were not recorded, limiting the generalizability of these results.”

Round 2

Reviewer 3 Report

It seems that the authors addressed all concerns raised by the reviewer properly.  They did acknowledge the limitations on this study; however well justified for what they could do since it is a retrospective human study.  The reviewer appreciate their scientific attitude.